# Detection of Thymoma Disease Using mRMR Feature Selection and Transformer Models

**DOI:** 10.3390/diagnostics14192169

**Published:** 2024-09-29

**Authors:** Mehmet Agar, Siyami Aydin, Muharrem Cakmak, Mustafa Koc, Mesut Togacar

**Affiliations:** 1Department of Thoracic Surgery, Faculty of Medicine, Firat University, 23119 Elazig, Turkey; siyamiaydin51@gmail.com (S.A.); g.c.dr.ckmk@gmail.com (M.C.); 2Department of Radiology, Faculty of Medicine, Firat University, 23119 Elazig, Turkey; mkoc44@yahoo.com; 3Department of Management Information Systems, Faculty of Economics and Administrative Sciences, Firat University, 23119 Elazig, Turkey; mtogacar@firat.edu.tr

**Keywords:** thymoma disease, thymoma detection, transformer model, feature fusion, feature selection

## Abstract

Background: Thymoma is a tumor that originates in the thymus gland, a part of the human body located behind the breastbone. It is a malignant disease that is rare in children but more common in adults and usually does not spread outside the thymus. The exact cause of thymic disease is not known, but it is thought to be more common in people infected with the EBV virus at an early age. Various surgical methods are used in clinical settings to treat thymoma. Expert opinion is very important in the diagnosis of the disease. Recently, next-generation technologies have become increasingly important in disease detection. Today’s early detection systems already use transformer models that are open to technological advances. Methods: What makes this study different is the use of transformer models instead of traditional deep learning models. The data used in this study were obtained from patients undergoing treatment at Fırat University, Department of Thoracic Surgery. The dataset consisted of two types of classes: thymoma disease images and non-thymoma disease images. The proposed approach consists of preprocessing, model training, feature extraction, feature set fusion between models, efficient feature selection, and classification. In the preprocessing step, unnecessary regions of the images were cropped, and the region of interest (ROI) technique was applied. Four types of transformer models (Deit3, Maxvit, Swin, and ViT) were used for model training. As a result of the training of the models, the feature sets obtained from the best three models were merged between the models (Deit3 and Swin, Deit3 and ViT, Deit3 and ViT, Swin and ViT, and Deit3 and Swin and ViT). The combined feature set of the model (Deit3 and ViT) that gave the best performance with fewer features was analyzed using the mRMR feature selection method. The SVM method was used in the classification process. Results: With the mRMR feature selection method, 100% overall accuracy was achieved with feature sets containing fewer features. The cross-validation technique was used to verify the overall accuracy of the proposed approach and 99.22% overall accuracy was achieved in the analysis with this technique. Conclusions: These findings emphasize the added value of the proposed approach in the detection of thymoma.

## 1. Introduction

Anterior mediastinal masses are lesions that require a multidisciplinary approach. Benign masses of thymus origin (thymolipomas, thymic cysts, etc.); neoplasms such as thymic hyperplasia, thymic carcinoma, and thymic carcinoid; germ cell tumors (15% of mediastinal masses); ectopic parathyroid tissue or adenoma (approximately 20% are ectopic); substernal goiter; lymphoma; hemangioma; and sarcoma are lesions located in the anterior mediastinum [1]. Although more than half of mediastinal masses are benign, approximately 59% of anterior mediastinal lesions are malignant [2]. The thymus is a lymphoid organ located in the anterior mediastinum that is part of the immune system in childhood and is expected to shrink and disappear in adulthood. While thymoma of thymus origin accounts for less than 1% of adult malignancies, it is the most common anterior mediastinal mass in adults. Thymomas are slow-growing neoplasms that arise from the epithelial cells of the thymus [3]. There is no gender or racial predisposition to thymoma, which usually presents in the fourth to sixth decade of life. A paraneoplastic syndrome is associated with thymoma in 30–50% of cases. The most commonly associated paraneoplastic syndrome is myasthenia gravis. While 30–50% of patients with thymoma have myasthenia gravis, only 10–30% of patients with myasthenia gravis have thymoma [4].

While a thymoma may be discovered incidentally in an asymptomatic patient, a symptomatic patient will present with symptoms related to the compression of the mass or a paraneoplastic syndrome. These symptoms range from non-specific findings such as chest pain, hemoptysis, dyspnea, cough, pleural/pericardial effusion, phrenic nerve palsy due to compression, dysphagia, hoarseness, Horner’s syndrome, and superior vena cava syndrome [5]. Computed tomography (CT) has an important role in the diagnosis of thymoma. With its high resolution and superiority in identifying soft tissues, it is very helpful in determining the size, content (cystic–solid), density, and relationship to adjacent tissues and vascular structures. Thymoma usually appears as a smooth and well circumscribed lesion during CT [6]. However, like other anterior mediastinal lesions, thymoma requires tissue sampling for definitive diagnosis [7]. The differential diagnosis of thymoma includes thymic carcinoid tumors, thymic cysts, lymphomas, germ cell tumors, ectopic parathyroid, goiters, and rarely, paragangliomas. Surgical resection is the first line of treatment for early-stage thymoma and is considered curative. For advanced, inoperable thymoma, treatment options such as radiation therapy and chemotherapy are used [8].

Recently, several AI-based studies on thymoma disease have emerged in the literature. Some of these studies, including Lei Yang et al. [9], proposed a deep learning approach that can successfully classify between stages of thymoma disease (stage I, stage II). The data used in their study consist of CT images. They used the ROI technique and then trained the data using the DenseNet model. They achieved an area under the curve (AUC) success of 77.3% in the two-step classification. Wei Liu et al. [10] performed analyses with deep learning models to classify subtypes of thymoma disease using CT image data. They applied the regional focusing technique with ROI to the images. Then, they extracted feature sets by training with the ResNet50 deep learning model. By applying the Lasso feature selection algorithm to the feature sets, they classified the efficient features using the multi-layer perceptron (MLP) method. The AUC success they achieved in their study was 99.8%. Zhenguo Liu et al. [11] used a deep learning approach to detect myasthenia gravis syndrome in thymoma disease. They used 3D-DenseNet-based multiple models for training CT images. In the classification process, they used methods such as Random Forest (RF), XGBoost, SVM, MLP, etc. The overall accuracy (Acc) performance of the model they used for training was 72.4%.

There are a limited number of recent publications on the detection and classification process of thymoma disease. The studies in the literature have traditionally used classical deep learning models and machine learning methods. Wei Liu et al. [10] used techniques and methods similar to our proposed approach in their work. Their use of the ROI technique and feature selection algorithm (lasso) improved the performance of their proposed approach. The fact that they relied on a deep learning model to train the dataset and stick to a single model may have limited the performance on potentially different or diverse images. Our proposed model uses next-generation transformers for analysis instead of classical deep learning models. It improves the overall performance with preprocessing steps (ROI) and post-processing steps (feature fusion, feature selection).

In this paper, a hybrid artificial intelligence (AI) approach is proposed for the successful detection of thymoma disease. Preprocessing steps are applied to facilitate the dataset training of the transformer models. The feature sets extracted from the fully connected layers (FC) of the transformer models are merged among the models themselves to obtain new feature sets. After the feature sets generated by the feature fusion technique are classified, the most efficient feature set is selected and processed using the Minimum Redundancy Maximum Relevance (mRMR) method. This method aims to achieve more efficient results with fewer features. A support vector machine (SVM) is preferred for the classification method. All these processing steps are expected to contribute positively to the classification process of thymoma disease. The innovative aspects and contributions of the proposed approach are as follows:Since the images were taken from the hospital environment, a region of interest (ROI) analysis was performed to eliminate unnecessary regions in the image. Thus, the approach aims to be more efficient when transferring images from real-time hospital environments to transformer models.The models were trained using transformers, a new-generation technology that has recently gained a reputation for producing more effective results than traditional deep learning models.Feature fusion between transformer models created feature sets with more efficient features, contributing to performance.The mRMR method was used to improve performance with fewer features but more efficient features by determining the most efficient combined feature set. This saves time and reduces hardware costs.An approach was proposed that can make objective decisions by avoiding possible disagreements between physicians and radiologists in the diagnosis of thymoma disease.A system was provided that provides feedback to many patients simultaneously in a short time using the proposed approach.

The remaining sections of the paper are organized as follows: Section 2 describes the dataset, techniques, methods, and models used in this study. Section 3 discusses the proposed feature selection-based transformer hybrid approach. Section 4 presents the experimental analyses. The last two sections are devoted to the discussion and conclusion, respectively.

## 2. Background

This section consists of subsections providing information about the dataset, techniques, methods, and models. Detailed information is provided on the data of thymoma patients obtained from Firat University Hospital (Elazig, Turkey). In addition, the approaches used in the proposed hybrid model are detailed in this section.

### 2.1. Dataset

The dataset is not open access and was created by specialized physicians. The images in the dataset were obtained from Fırat University Research Hospital. A total of 298 patients, including 128 patients with radiologically and pathologically diagnosed thymoma and 170 patients with non-thymoma anterior mediastinal lesions between 2011 and 2024, were included in the retrospectively designed study. A total of 768 sections were included in the study, with an average of three chest CT images from the thymoma group and 2.25 from the non-thymoma group. Of the patients diagnosed with thymoma, 91 were male and 37 were female. The mean age was 54.6 ± 16.8 years. The mean diameter of the thymoma on CT was 4.8 ± 1.9 cm. The most common stage was 2a according to the Masaoka staging system and the most common histopathological type was B2 according to the World Health Organization (WHO) classification (see Table 1). In addition, myasthenia gravis was detected in 39 patients with thymoma presenting with various symptoms (see Table 1). In the non-thymoma group, 108 patients were male and 62 were female. The mean age was 56.5 ± 15.6 years. The non-thymoma group included patients with thymic hyperplasia, lymphoma, thymic cyst, thymic carcinoma, germ cell tumors, and ectopic parathyroid tumors (see Table 2).

The dataset consists of two classes. The number of images with thymoma disease is 384. The number of images without thymoma is 384. The images in the dataset are equally classified and consist of a total of 768 images. A sample subset of images belonging to the types of the dataset is shown in Figure 1.

### 2.2. Preprocessing Step

The goal of the preprocessing steps is to transform the data into a more efficient format. For this purpose, operations such as data augmentation, data cleaning, data merging, normalization, regional focusing or cropping of unnecessary fields, etc., are known as preprocessing steps. These choices may vary depending on the type of dataset, size, data size, or analysis choices. Since the dataset images in this study are of variable resolution and there are many redundant regions in the image, we applied two techniques: data cropping and region of interest (ROI).

For variable-resolution images, the data clipping technique removes unnecessary regions and emphasizes the image area that should be in focus. This ensures a successful training result for training AI models. Each image in the dataset used in this study was of variable resolution, and each image was converted (cropped) to 224 × 224 resolutions in a center–midpoint format. All these operations were performed using the Python 4.0 language Torch and Numpy libraries. The image resolution of 224 × 224 was chosen because it is the same as the input resolution of the transform models. Example transform images showing this process step are shown in Figure 2. After the data cropping technique, the ROI technique was applied to each image using the Python language, and the sample image set obtained is shown in Figure 3.

### 2.3. SVM Classifier

The SVM is a machine learning method that is widely used for regression and classification tasks. It is a method that is often preferred in binary and multiple classification processes and can perform successful classification. This method places the features extracted from the data on a virtual coordinate plane. It then creates a boundary region that divides the dataset into two classes. This region is called the hyperplane. The maximum boundary area is determined by the optimization method. In the final stage, the classification probability is calculated using the following equations, and the classification is performed by the SVM. The binary classification design of the SVM is shown in Figure 4.

The objective of Equation (1) is to reduce the issues that arise during the classification process. In Equation (3), $X_{i}$ and $Y_{i}\;$ denote the features. W is half the width of the margin, while b is a bias term that determines the location of the decision boundary. This equation provides an estimate to identify the class of the input data [12,13,14].
(1)$$u=\vec{w}\cdot \vec{x}-b$$
(2)$$\;\frac{1}{2}{\left\|\vec{w}\right\|}^{2}$$
(3)$$y_{i}(\vec{w}\cdot {\vec{x}}_{i}-b)\geq 1,\forall i$$

The important parameters and values of the SVM method for the proposed hybrid model are given in Table 3. These values are the default values of the method.

### 2.4. Feature Selection Method: mRMR

In AI-driven models or approaches, feature selection techniques are used to improve classification performance. The goal is to increase accuracy by identifying and emphasizing the most effective features from the set derived from the model layer. Essentially, these methods allow the creation of a refined subset of features by selecting the most relevant ones from the original set [15,16]. For this study, the minimum redundancy maximum relevance (mRMR) method [17,18,19,20], which can perform effective selection in datasets containing binary types, was preferred. In the literature, many researchers have stated that this selection is successful in the analysis. The Feature Selection Methods tool of the MATLAB interface was used for the analysis.

The mRMR algorithm is a filtering technique used to select the most appropriate features by analyzing the feature sets derived from the input data. It works well for binary classification tasks. The algorithm assigns a ranking to the features and attempts to establish relationships between these ranked features using its own method. It also evaluates these relationships with a scoring system, where a higher score indicates a stronger relationship. As a result, mRMR identifies the least relevant features as residuals and the most relevant features as matches. In essence, it focuses on identifying similarities between features [21,22].

Here, the maximum fitness (W) and minimum redundancy (V) values are calculated by the mRMR algorithm. Finally, a score table is generated according to Equation (4) and the score of each feature is calculated. A ranking is conducted between the feature scores.
(4)max (VW)

### 2.5. Transformer Models

Recently, deep learning models have gained significant attention in various domains. Convolutional Neural Networks (CNNs) have particularly advanced the classification process in autonomous medical image analysis applications. However, a major limitation of CNN models is their inefficiency in learning long-range dependencies due to their localized receptive fields, which limits their performance in vision-related tasks. In contrast, vision transformers (ViTs) excel at handling long-range information and have become a prominent topic in computer vision. Due to their superior attention mechanisms, ViT models often outperform CNNs in critical health-related computer vision tasks where decisions can have life-threatening consequences. Despite being simpler in terms of parameter density compared to CNNs, ViT models can achieve better performance. Their architectures are well suited for applications such as object detection, image processing, and classification [23,24,25,26].

Four transformer models were used for this study. The common feature of the transformer models is that their input resolution is 224 × 224. This means that each image obtained in the preprocessing step had the same resolution. When the general structure of the transformer models is examined, it performs its operations in six stages. Of course, depending on the feature structure of the models, there may be different structures/layers/blocks outside the stages [25,27,28]. These process stages are shown in Figure 5.

The first stage is input embedding: This stage deals with the reception of image data. The images are divided into patches for processing. First, each image is segmented into a sequence of two-dimensional patches. These patches can have specific resolutions, such as 16 × 16 or 24 × 24 pixels.The next stage is patch embedding. During this stage, each patch is processed through an embedding layer. At the end of this stage, each patch is transformed into a vector representation.The third stage is position embedding. At this point, the order of the patches is determined and position embeddings are incorporated accordingly. This allows the model to easily determine the position of each patch.The fourth stage contains the transformer encoder blocks. The ViT architecture includes several transformer encoder blocks, each of which consists of two main components. The first component is Multi-head Self-Attention (MHSA), which is responsible for querying and modeling correlations between image patches. The second component is a feed-forward neural network (FFNN), which extracts and processes features from each patch and then applies the necessary transformations for the next stage.The fifth stage is the transformer encoder stack. During this phase, multiple encoder blocks are stacked, allowing for deeper feature extraction and learning of more complex correlations.The final stage is the output layer, which is common to all models. This layer aggregates the data processed by the transformer encoder stack. It is typically used for classification tasks and serves as the final layer in CNN models. For classification purposes in ViT models, activation functions such as softmax are commonly used.

By passing input images through convolutional layers, ViT effectively captures intricate patterns and contexts within images. The processing improves visual performance by breaking images into manageable segments and using self-attention modules. Essentially, ViT processes the input image by dividing it into fixed-size patches. These patches are then flattened into 1D vectors. Each patch is processed by a transform encoder, ensuring that all contextual dependencies within the image are maintained throughout the processing. Segmenting the image into patches allows the transform model to preserve spatial interactions while simplifying the computational complexity of handling large images, thus facilitating the training process [23,29]. Four variations of the transformer model were selected for this study. These models included the following:Deit3 base patch16-224: DeiT (Data-efficient Image Transformer) was developed to improve the efficiency of transform models in image processing. It is a type of DeiT3 series, known as the third generation version of a ViT-based model. The model receives input images with a resolution of 224 × 224 pixels. The images are divided into small patches of 16 × 16 pixels. These patches allow the model to learn local and distant features on the image. The model generally contains fewer transformer encoder blocks and has fewer parameters than the DeiT model types. Because it is a smaller model, it uses less memory and can compute faster. This makes it easier to use, especially in computing environments with limited hardware resources. Another advantage of this model is that it can capture important details in complex images. The DeiT3 series aims to achieve high accuracy with less data [30,31].MaxViT base-tf-224: MaxViT (Maximum Vision Transformer) is a ViT-based model designed for image classification and other computer vision tasks. There are several types of MaxViT, the simplest and most basic model is ‘maxvit base-tf-224’. This model takes input images with a resolution of 224 × 224 pixels. The images are not processed as a whole but are divided into parts of certain scales according to the internal structure of the model. MaxViT is generally optimized for computation and memory. The model is often preferred for image classification, object recognition, and segmentation. For the model to perform well, the dataset must be large and diverse [32,33].Swin base patch4 window7-224: Swin (Shifted Window) Transformer is an extended version of the ViT models. Swin Transformer uses window-based attention mechanisms to learn both local and global contexts, enabling the processing of multi-scale features. This model can process input images with a resolution of 224 × 224 pixels. The term Patch4 indicates that the model divides each image into 4 × 4 pixel patches. The term Windows7 refers to the window size that the model uses for attention computations, which covers an area of 7 × 7 pixels during each attention operation, making it possible to learn both local and large-scale contexts. This model is preferred for image classification, segmentation, and object recognition [34].ViT base patch16-224: The model accepts images with an input resolution of 224 × 224 pixels and processes them by dividing each image into 16 × 16 pixel patches. This version is a simplified version of the ViT-large model. It has fewer transform encoder blocks and a generally smaller number of parameters. It also uses less memory [28,35,36].

### 2.6. Proposed Hybrid Approach

The proposed approach consists of artificial intelligence-based methods and models that can find the diagnosis of thymoma disease in real-time by analyzing CT images. To achieve the goal of this study, ViT models, which have recently assumed a more effective role in image processing and efficient feature extraction than CNN models, took the main role. In order to add a more innovative aspect to these models and increase the diagnostic success of the disease, we synthesized the model with hybrid approaches. The proposed hybrid model consists of six main steps in terms of design. These steps are preprocessing, model training, feature extraction, feature set fusion between models, feature selection, and classification.

In the preprocessing step, image enhancement, image cropping, and highlighting of necessary regions in the image are performed. The goal of this step is to improve the training performance of the ViT models in the next step and to extract more efficient features from disease images. The raw data acquired in hospital environments have different resolutions. To take advantage of this, each image is preprocessed. In the preprocessing step, each variable-resolution image is cropped using the center-middle points. Image normalization is performed and then regional focusing is performed using the ROI technique. The resolution of the images is fixed, and all are set to a 224 × 224 pixel resolution format. The 224 × 224 resolution was chosen because the ViT models process the image at this resolution. Therefore, the input resolution of the models is also 224 × 224. This avoids additional costs for the models.

ViT models (Deit3/base patch16, MaxViT/base-tf, Swin/base patch4, and ViT/base patch16) were used in the model training step. The model architectures do not contain too many parameters and are not as complex as their counterparts. For this reason, they may be easier to install on devices with fewer hardware requirements. For this study, multiple ViT models were used to enrich the feature sets without relying on a single model. Each model takes an input image with a resolution of 224 × 224 pixels. Therefore, having alternative models in a hybrid approach can positively contribute to different types of dataset images. Feature sets are extracted from the fully connected (FC) layer, which is located before the layer where the classification of the four ViT models takes place. The DeiT3, MaxViT, and ViT models each yield 768 feature sets, and the Swin model yields 1024 feature sets due to its different architecture. SVM is used to measure the performance of the feature sets associated with the models. Given the performance of the models, the feature sets of the top three models are determined. The goal is to save time and reduce costs. The next step is to combine the model-based feature sets and perform the analysis. For this study, the top three ViT models were Deit3, Swin, and ViT. This may vary for different datasets. For this study, the feature sets of the models were combined with each other. Thus (Deit3 and Swin, Deit3 and ViT, Swin and ViT, and Deit3 and Swin and ViT) new feature sets with four possibilities were obtained. The SVM method was then used to measure the performance of the combined model-based feature sets.

Among the model-based combined feature sets, the best-performing set is passed to the next stage. Thus, a general-to-specific process is performed so that cost and time factors can be favorable. The best-performing combined feature set is processed with the mRMR method. In this step, sets with 50, 100, 200, …, 1000 features are determined by the mRMR method and classified by the SVM method by considering the best features, respectively. So, is a better performance result obtained with sets containing fewer features? This can be seen in the last step. Experimental studies have shown that this also contributes, and the general design of the proposed hybrid approach is shown in Figure 6.

## 3. Experimental Analysis

MATLAB 2024 was used for the analysis, which included feature fusion and feature selection methods, as well as the classification process using machine learning. The analysis was performed using computational resources, including an Intel Core i7 processor at 3.40 GHz, 32 GB of RAM, and a graphics card with 10 GB of memory. The Jupyter Notebook interface was used to train the ViT models. In addition, the ViT models were analyzed on Google Colab servers.

A confusion matrix is often used in classification processes [37]. The confusion matrix was used to validate the analyses performed in this study. The metrics of the confusion matrix are calculated using the equations provided in Equations (5)–(9) [38,39,40]. The terms used in the equations include positive (P), negative (N), true (T), and false (F). The metric abbreviations in the equations are as follows: accuracy (Acc), f-score (F-Scr), specificity (Sp), sensitivity (Se), and precision (Pre). The Acc metric is commonly used in measurement analysis and works well with balanced datasets. The F-Scr metric is particularly useful in cases where there is an imbalance in the class distribution [41,42].
(5)$$\mathrm{Acc}=\;\frac{TP+TN}{TP+TN+FP+FN}$$
(6)$$F\text{-}\mathrm{Scr}=\frac{2\times TP}{2\times TP+FP+FN}$$
(7)$$\mathrm{Sp}=\;\frac{TN}{TN+FP}$$
(8)$$\mathrm{Se}=\;\frac{TP}{TP+FN}$$
(9)$$\mathrm{Pre}=\frac{TP}{TP+FP}$$

The preferred parameters and their values for the ViT models used in this study are detailed in Table 4. The training and test data contain different images from the same patient. Experimental analyses with the proposed approach consist of three steps.

The first step is to refine the dataset to make it more useful before training the transform models. For each image, the Python code ’ResizedCrop(224)’ was used to reconstruct the resolution from the center of the image to 224 × 224. Then, the ROI technique, supported by the Pandas and Numpy libraries and designed in Python, was applied to the images to foreground the desired regions. An example image subset of this step is shown in Figure 2 and Figure 3.

In the second step, the data processed in the preprocessing step were trained with transformer models (Deit3/base patch16, MaxViT/base-tf, Swin/base patch4, and ViT/base patch16). The goal was to identify the three best models that could provide the best feature set using model diversity. The classification performance of the transformer models was also measured. In the experimental analysis, 80% of the dataset was divided into training data and 20% into test data. The ‘linear’ layer was used in the classification process of the transformer models. The overall accuracy performance graphs of the models are shown in Figure 7. The confusion matrices of the models are shown in Figure 8 and the metric results obtained from the confusion matrices are given in Table 5.

When analyzing Table 5, the Swin model gave the best performance and achieved 96.10% accuracy. The MaxViT model gave the worst performance and achieved 88.31% accuracy. The use of different ViT models in the proposed approach may increase the time cost, but it allows the formation of feature sets that can increase performance success. It is stated that the three best-performing models in the proposed approach will proceed to the next step. Since the DeiT3, Swin, and ViT models performed better in this study, the feature sets of these models were used in the next step.

In the second step, the feature sets were merged between the models by considering the top three model performances as shown in Table 5. The goal of the second step was to combine the feature sets obtained from the models DeiT3 and Swin, DeiT3 and ViT, Swin and ViT, and DeiT3 and Swin and ViT) to see if this improves the performance. This was performed by extracting feature sets from the fully connected layer (the last layer before classification) of the models. The DeiT3 model yielded 768 feature sets, the Swin model yielded 1024 feature sets, and the ViT model yielded 768 feature sets. The feature sets of these models were then merged between the models. Combining the feature sets of the DeiT3 and Swin models resulted in 1792 feature sets. Combining the feature sets of DeiT3 and ViT models resulted in a feature set of 1536 features. A total of 1792 feature sets were created by combining the feature sets of Swin and ViT models. Finally, 2560 feature sets were created by combining the feature sets of DeiT3 and Swin and ViT models. The statistical information and the letter symbol representing the merged models after the merging process was completed are shown in Table 6.

The SVM method was then used to measure the performance of the four combined feature sets. In the analyses performed in this step, the test data were set to 20% and a cross-validation technique was used to confirm the validity of the performance (k = 10 was chosen). The confusion matrices obtained from the test data as a result of the analysis are shown in Figure 9. The confusion matrices obtained with the cross-validation technique are shown in Figure 10. The metric results of the confusion matrices in Figure 9 and Figure 10 are given in Table 7. The inter-model feature fusion approach and classification with SVM showed 100% overall success performance as a result of the analyses performed by separating the dataset into training and test.

However, to verify the accuracy of this performance, the dataset was processed with a cross-validation technique (k = 10) and classified with the SVM method. As a result of this process, the Y model achieved an overall accuracy of 98.95% and the W-V-Z models achieved an overall accuracy of 99.08%. The cross-validation technique confirms the performance of the proposed approach.

The third step of the experimental analysis is to determine whether the feature selection algorithm can perform better with fewer features. We sought an answer to this question in the third step. To find the best performing combined model (W, V, Y, Z), Table 6 and Table 7 were examined. In these tables, the accuracy and the number of features were the reference. In Table 7, the ‘W, V, Y, Z’ models performed the same (100%) on the test data. However, in the cross-validation analysis, the ‘W, V, Z’ models showed the best success with an overall accuracy of 99.08%. Table 6 was analyzed to make a logical choice among the ‘W, V, Z’ models. In Table 6, it is observed that the ‘V’ model with 1536 features performs as well as the ‘W’ and ‘Z’ models with fewer features. The feature set obtained with the ‘V’ (DeiT3 and ViT) model with fewer features is used in the third step.

In the third step of the experimental analysis, the feature set generated with the ‘V’ model (DeiT3 and ViT) was analyzed using the mRMR method. For the mRMR method, the feature selection tool of the MATLAB application was used. The example of the feature ranking obtained as a result of this application is shown in Table 8. Here, the best 50, 100, 200, 300, 400, 500, 750, and 1000 feature sets were identified in the ‘V’ model-based feature set. These feature sets were then classified using the SVM method. In the classification process, as in the other steps, the test data were separated by 20% and the analysis was performed using the cross-validation technique (k = 10 was chosen).

The confusion matrices obtained from the analysis of this step with test data are shown in Figure 11. Similarly, the confusion matrices obtained from the analysis of the data separated by the cross-validation technique are shown in Figure 12. The overall accuracy performances of the features of the ‘V’ feature set selected by the mRMR method are given in Table 9. Analyzing Table 9, when considering the overall accuracy performance obtained as a result of separating the data with the ‘test’ and ‘cross-validation’ techniques, the top 500 features gave the best performance. While the Top 500 feature set achieved 100% overall accuracy with the test data, it achieved 99.22% overall accuracy in the analysis performed with the cross-validation data.

The top 200 and top 400 feature sets also performed close to the top 500. The other feature sets also performed close to each other. In short, the feature selection method contributes to the performance of the proposed approach with fewer features.

## 4. Discussion

The thymus is part of the immune system and is responsible for the development of T lymphocytes. Thymoma is a slow-growing tumor that usually does not spread beyond the thymus gland. Radiologic imaging is usually used to diagnose thymoma. Treatment depends on the patient’s overall health and the severity of symptoms. Depending on the stage of the disease, surgery may be preferred. Early diagnosis is important so that these treatments can be started at the mildest stage. Today, new-generation technology-based applications are becoming more popular than traditional applications. The fact that artificial intelligence has recently made a name for itself in various fields reinforces this. This study proposes an AI-based hybrid model for the early diagnosis of thymoma disease.

The proposed hybrid model consists of preprocessing steps, model training, and post-processing steps. These three steps contributed to the disease diagnosis in the experimental analysis. In the preprocessing step, the fixed resolution of the images, the cropping of unnecessary image areas, and the highlighting of specific regions in the image saved time and facilitated the training of the transformer models. ViTs offer advantages over CNN models, such as patching the image and efficiently using attention modules for each patch, a better understanding of contextual information, and better adaptation to larger or more complex datasets. Although the limited number of datasets in the analysis of this study is a disadvantage for the proposed approach, the use of ‘base’-based transformer models after the preprocessing step helped to achieve high-performance results with fewer data.

We used four types of ViT models in the training part of the study, and this choice has more advantages. Of course, there are costs such as high memory consumption and high computational power (drawbacks). However, for different types of data, the variety of models allows the low performance of one to be compensated by the high performance of another. This does not affect the overall performance too much. To minimize the time loss that may occur here, we proceeded to the next step by selecting the three best-performing models among the four models we trained. Another aspect that makes the proposed approach different is that the feature sets of the three ViT models were combined among themselves. Thus, we have seen in the experimental analysis that the combined feature sets contribute to the performance. Here, we achieved an overall accuracy of up to 100%. In the next step, we used the best-performing feature set and analyzed it with the mRMR method. The mRMR method allowed us to achieve more effective performance with fewer features. We realized this in the experimental analysis in the last step. We achieved 100% overall accuracy in the analysis performed with test data and 99.22% overall accuracy in the analysis performed with cross-validation. Here, the cross-validation technique was used to train the performance of our proposed model, as it distributes the training and testing part of the dataset more evenly. Another reason for the high overall performance is that we used 80% of the dataset as training data. Considering the overall performance, our proposed approach was successful in detecting the disease. The disadvantages of the proposed approach are that it is not an end-to-end model and the time cost.

We have summarized similar studies in the literature in Table 10. The number of AI-based studies for this disease is limited. We have included detailed information about the studies in Table 10 in the ’Related Works’ section. The most important feature that distinguished us from the other studies in Table 10 was the use of transformer models, which are new-generation models, in the analysis. Other studies analyzed traditional CNN models. Wei Liu et al. [10] performed their analysis with similar approaches to our study. However, the fact that they did not use transformer models, did not use the feature fusion technique, and most importantly stuck to a single CNN (ResNet) model limited their performance. The low performance of the other studies [9,11] was due to the fact that they analyzed with a single CNN model and did not contribute to the preprocessing/post-processing steps with innovative approaches.

## 5. Conclusions

The thymus gland, an essential organ of the immune system, is larger in childhood and shrinks to a size that is no longer visible in adolescence. Since cancer cells can be seen in many organs, they can also be seen in the thymus, although rarely. The most common type of thymic cancer is known as a thymoma tumor. As in the diagnosis of any disease, early diagnosis of thymoma is important. This paper aims to evaluate the contribution of AI approaches in the diagnosis of thymoma disease. The proposed approach consists of image preprocessing steps, training of transform models, feature set fusion between models, feature selection application, and classification processes. Keeping unnecessary areas in the background in the preprocessing steps was effective in the efficient selection of features extracted from the transform models. Feature set fusion across models also contributed to the overall performance of the proposed approach. The experimental analysis proves this. Finally, it was observed that the mRMR method, which has recently made a name for itself by performing effective feature selection, performs better with fewer features. In this study, the proposed approach achieved 100% overall accuracy with the test data and 99.22% overall accuracy with the cross-validation technique.

In the next study, the proposed approach will be applied to other disease diagnoses, provided that it offers different alternatives. In this context, meta-heuristics will be used instead of feature selection methods to select efficient features. In addition, other types of ViT models will be used and analyses will be performed with models that provide efficient results.

## Figures and Tables

**Figure 1 diagnostics-14-02169-f001:**
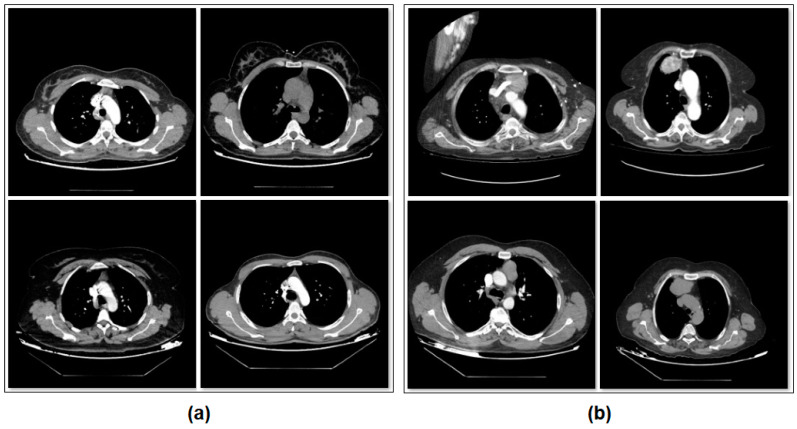
Sample images from the dataset categories: (**a**) patients without thymoma, and (**b**) patients with thymoma.

**Figure 2 diagnostics-14-02169-f002:**
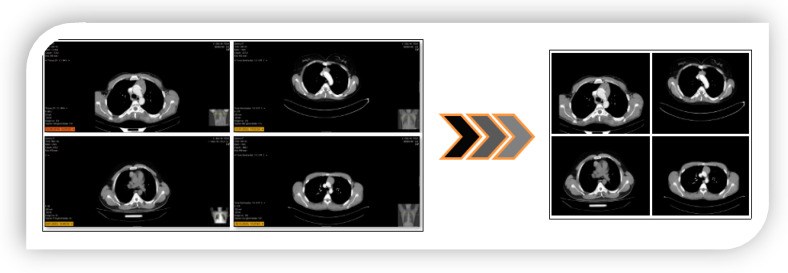
Application of data cropping technique to original data: example subset of images.

**Figure 3 diagnostics-14-02169-f003:**
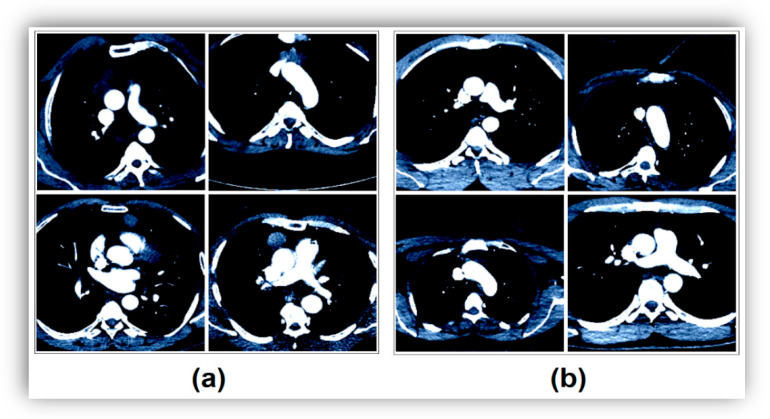
Application of the ROI technique after data cropping: (**a**) CT images without thymoma disease, and (**b**) CT images with thymoma disease.

**Figure 4 diagnostics-14-02169-f004:**
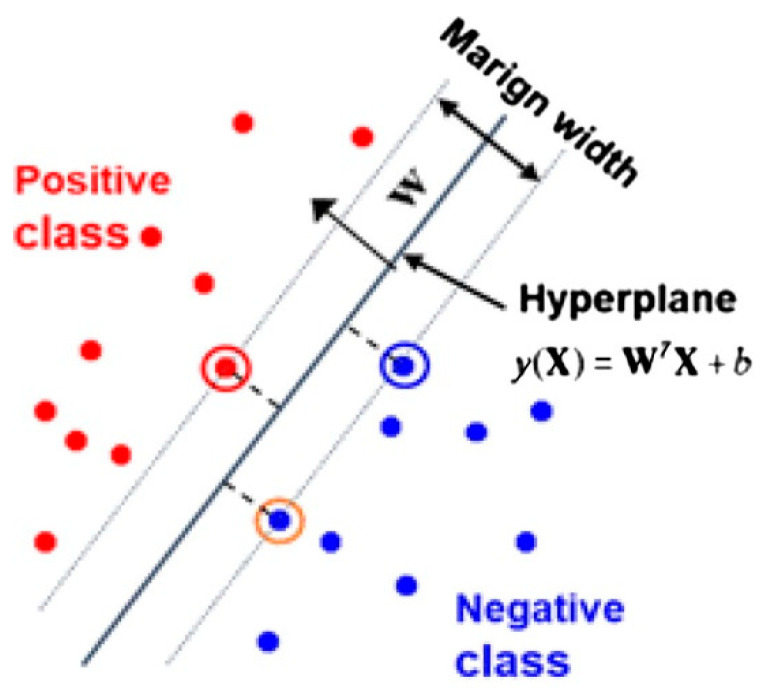
Binary-type classification by SVM [12].

**Figure 5 diagnostics-14-02169-f005:**
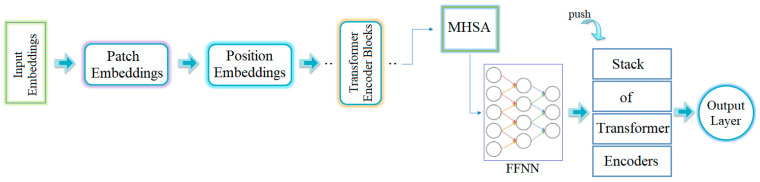
Process stage of the transformer models.

**Figure 6 diagnostics-14-02169-f006:**
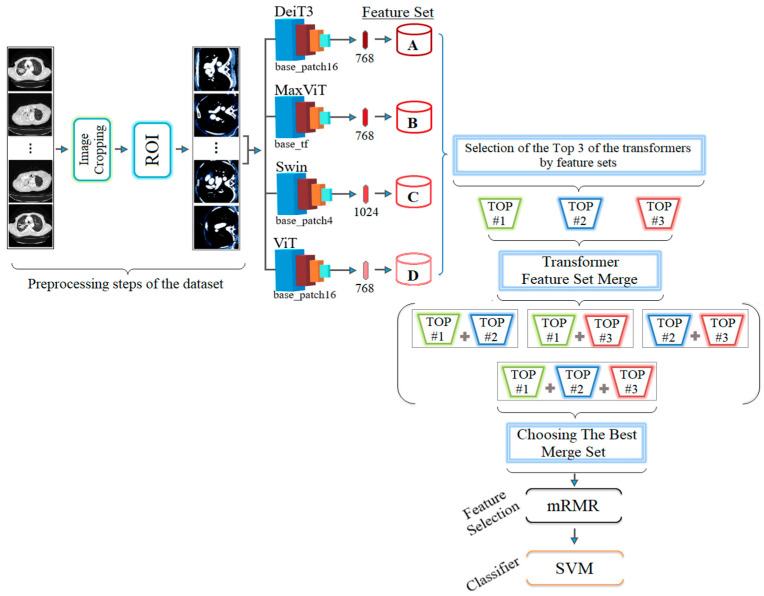
General design of the proposed hybrid model.

**Figure 7 diagnostics-14-02169-f007:**
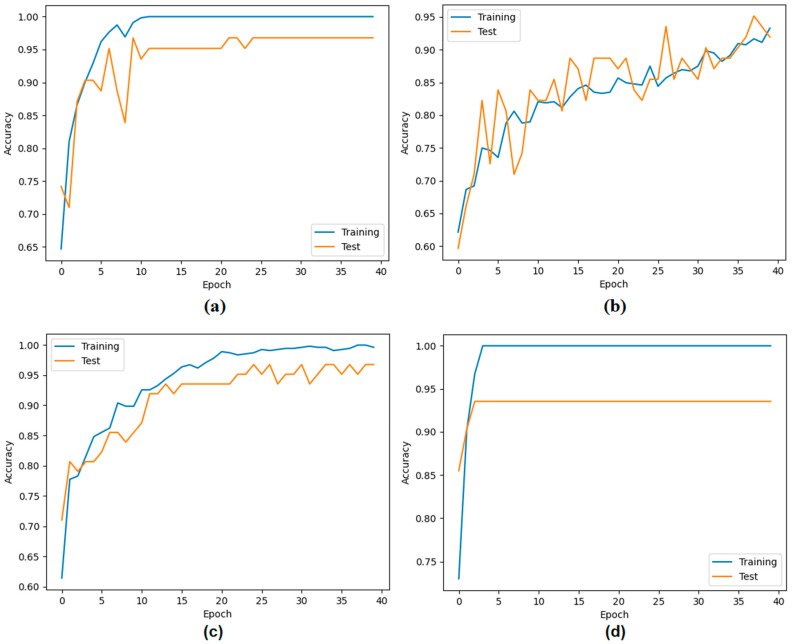
Classification performances of ViT models: (**a**) Deit3/base patch16, (**b**) MaxViT/base-tf, (**c**) Swin/base patch4, and (**d**) ViT/base patch16.

**Figure 8 diagnostics-14-02169-f008:**
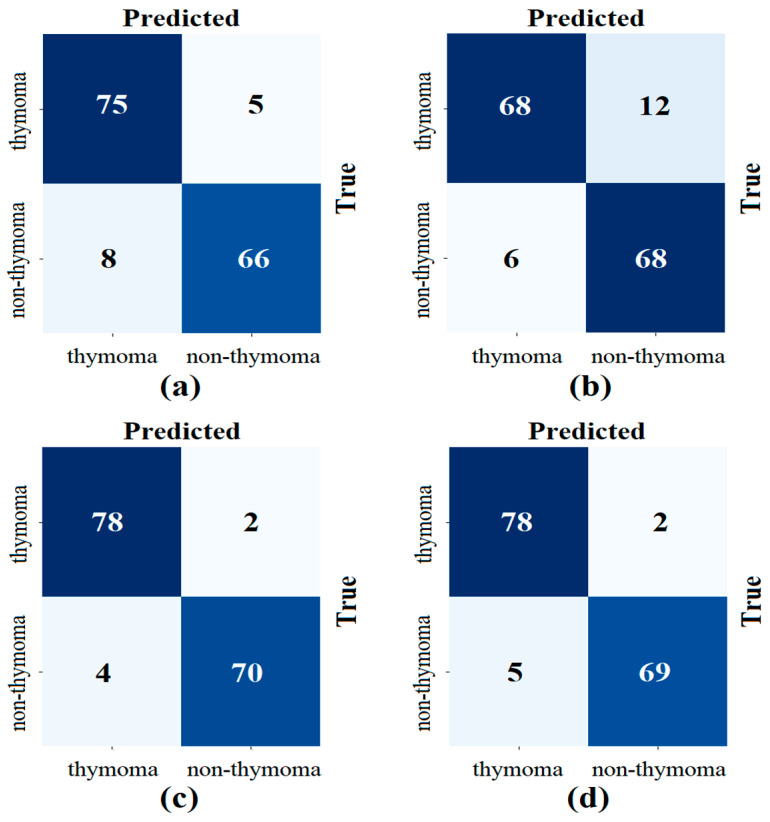
Confusion matrices obtained in the classification process of transformer models; (**a**) Deit3/base patch16, (**b**) MaxViT/base-tf, (**c**) Swin/base patch4, and (**d**) ViT/base patch16.

**Figure 9 diagnostics-14-02169-f009:**
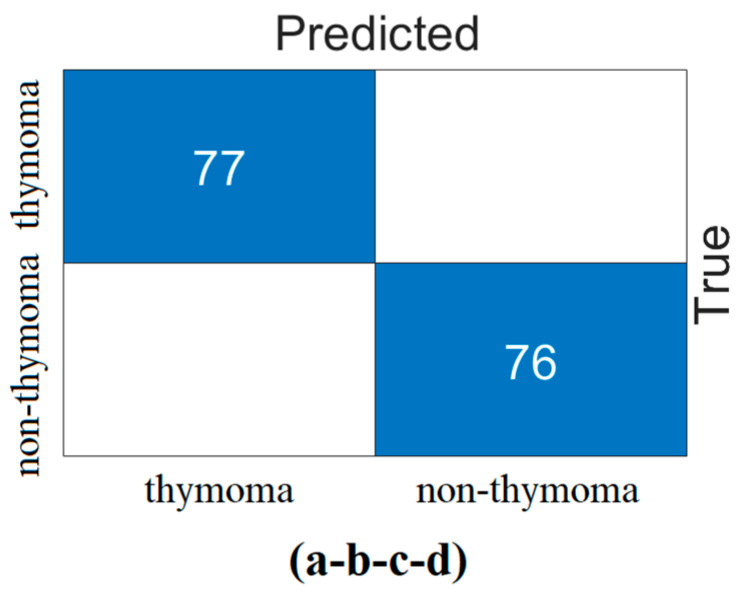
Confusion matrices obtained by combining the feature sets between the models (training/test rate: 0.8/0.2—classifier: SVM): (**a**) DeiT3 and Swin (W), (**b**) DeiT3 and ViT (V), (**c**) Swin and ViT (Y), and (**d**) DeiT3 and Swin and ViT (Z).

**Figure 10 diagnostics-14-02169-f010:**
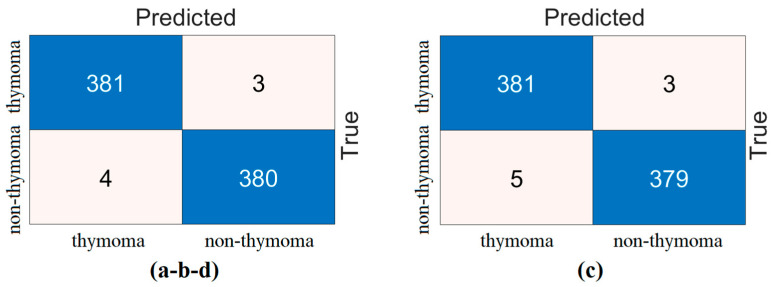
Confusion matrices obtained by combining the feature sets between the models (cross validation: k = 10 and classifier: SVM): (**a**) DeiT3 and Swin (W), (**b**) DeiT3 and ViT (V), (**c**) Swin and ViT (Y), and (**d**) DeiT3 and Swin and ViT (Z).

**Figure 11 diagnostics-14-02169-f011:**
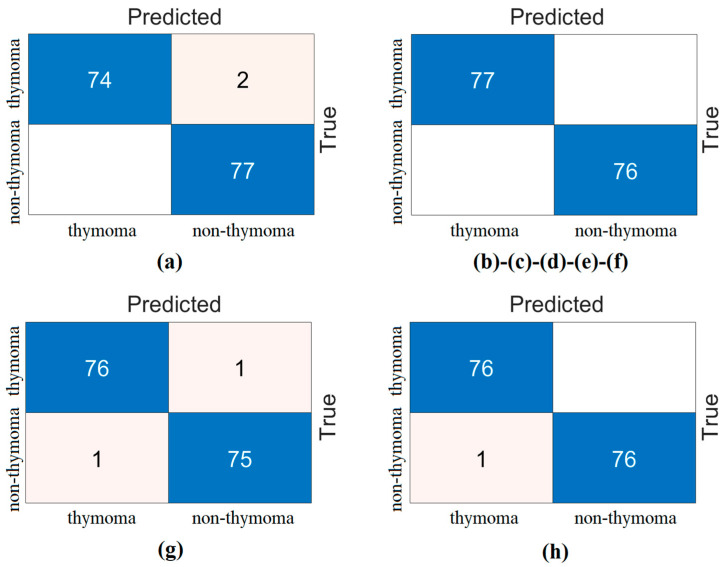
Confusion matrices obtained by SVM classification of the best features selected by mRMR feature selection method (train rate: 0.8, test rate: 0.2): (**a**) the top 50 features, (**b**) the top 100 features, (**c**) the top 200 features, (**d**) the top 300 features, (**e**) the top 400 features, (**f**) the top 500 features, (**g**) the top 750 features, and (**h**) the top 1000 features.

**Figure 12 diagnostics-14-02169-f012:**
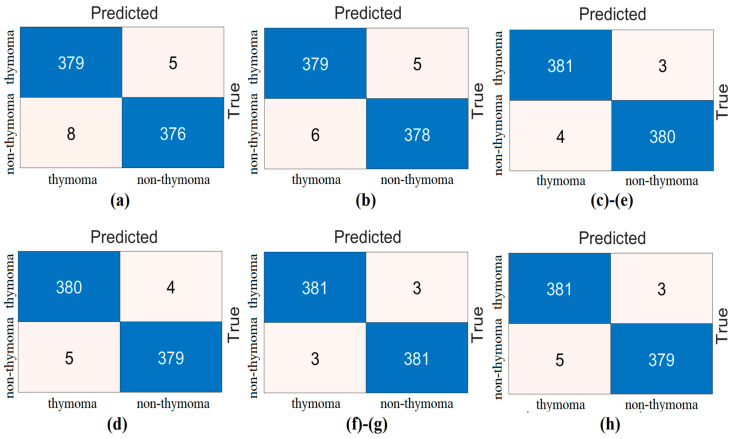
Confusion matrices obtained by SVM classification of the best features selected by mRMR feature selection method (cross validation/k = 10): (**a**) the top 50 features, (**b**) the top 100 features, (**c**) the top 200 features, (**d**) the top 300 features, (**e**) the top 400 features, (**f**) the top 500 features, (**g**) the top 750 features, and (**h**) the top 1000 features.

**Table 1 diagnostics-14-02169-t001:** General patient information of the thymoma group.

Gender	
Male	91 (%71.1)
Female	37 (%28.9)
Age (year, mean ± SD)	54.6 ± 16.8
CT size (year, mean ± SD)	4.8 ± 1.9
Masaoka Staging System	
Stage 1	21 (%16.4)
Stage 2a	58 (%45.4)
Stage 2b	38 (%29.7)
Stage 3	11 (%8.5)
WHO histologic classification	
A	23 (%17.9)
AB	21 (%16.4)
B1	19 (%14.9)
B2	34 (%26.6)
B3	31 (%24.2)
Symptoms	
Cough	
Yes	27 (%21.1)
No	101 (%78.9)
Chest pain	
Yes	40 (%31.2)
No	88 (%68.8)
Chest Distress	
Yes	37 (%28.9)
No	91 (%71.1)
Myasthenia Gravis	
Yes	39 (%30.5)
No	89 (%69.5)

**Table 2 diagnostics-14-02169-t002:** General patient information of the non-thymoma group.

Gender	
Male	108 (%63.5)
Female	62 (%36.5)
Age (year, mean ± SD)	56.5 ± 15.6
Diseases of the non-thymoma group	
Thymic hyperplasia	51 (%30)
Lymphoma	45 (%26.5)
Thymic cyst	39 (%22.9)
Thymic carcinoma	18 (%10.6)
Germ cell tumour	11 (%6.5)
Ectopic parathyroid-thyroid	6 (%3.5)

**Table 3 diagnostics-14-02169-t003:** Hyperparameters used in machine learning method.

SVM Parameter	Parameter Value/Choice
Kernel function	Cubic
Kernel scale	Auto
Box constraint level	1
Multiclass method	One-vs-One

**Table 4 diagnostics-14-02169-t004:** Parameters and values of ViTs used in the proposed approach.

Model	Parameter	Preference/Value
DeiT3	Loss function	Cross Entropy
	Learning rate	1 × 10^−4^
MaxViT	Optimization	SGD
	Classifier	Linear
Swin	Epoch	40
	Mini-batch	2
ViT	Training rate: testing rate	0.8:0.2

**Table 5 diagnostics-14-02169-t005:** Metric results from the confusion matrix of transformer models (%).

Transformer Model	Features	Class	Se	Sp	Pre	F-Scr	Acc	TOP #N
Deit3/base patch16	768	thymoma	93.75	89.18	90.36	92.02	91.55	#3
		non-thymoma	89.18	93.75	92.95	91.03		
MaxViT/base-tf	768	thymoma	85.0	91.89	91.89	88.31	88.31	#4
		non-thymoma	91.89	85.0	85.0	88.31		
Swin/base patch4	1024	thymoma	97.50	94.59	95.12	96.29	96.10	#1
		non-thymoma	94.59	97.50	97.22	95.89		
ViT/base patch16	768	thymoma	97.50	93.24	93.97	95.70	95.45	#2
		non-thymoma	93.24	97.50	97.18	95.17		

**Table 6 diagnostics-14-02169-t006:** The number of features of models combined using feature sets and the letter symbols they represent.

Representing Letter Symbol	Models with Combined Features	Feature Counts
W	DeiT3 and Swin	1792
V	DeiT3 and ViT	1536
Y	Swin and ViT	1792
Z	DeiT3 and Swin and ViT	2560

**Table 7 diagnostics-14-02169-t007:** Metric results of confusion matrices obtained from the analysis of model-based fused feature sets.

Feature Merging between Models	Dataset	Class	Se	Pre	F-Scr	Acc
W, V, Y, Z	train/test rate 0.8:0.2	thymoma	100	100	100	100
		non-thymoma	100	100	100	
W, V, Z	Cross-validation (k = 10)	thymoma	99.21	98.96	99.09	99.08
		non-thymoma	98.95	99.21	99.09	
Y		thymoma	99.21	98.70	98.96	98.95
		non-thymoma	98.69	99.21	98.95	

**Table 8 diagnostics-14-02169-t008:** Feature ranking subset realized with mRMR feature selection.

No	Column Number	Score
1	124	0.6145
2	785	0.6118
3	1308	0.2060
4	68	0.1921
5	262	0.1714
6	1009	0.0821
⋮		
1535	1400	0.0001
1536	606	0.0000

**Table 9 diagnostics-14-02169-t009:** The overall accuracy of the mRMR method for the classification of selected features.

Combined Feature Set	Machine Learning	Top Features	Acc (%) (Test: 0.2)	Acc (%) (Cross val./k = 10)
‘V’ 1536 features	SVM	50	98.69	98.31
		100	100	98.57
		200	100	99.09
		300	100	98.83
		400	100	99.09
		500	100	99.22
		750	98.69	99.22
		1000	99.35	98.96

**Table 10 diagnostics-14-02169-t010:** Studies in the AI-based literature on thymoma disease.

Article	Year	Model/Method	Result
Lei Yang et al. [9]	2020	DenseNet and ROI	AUC: 0.730
Wei Liu et al. [10]	2024	ResNet50 and ROI and Lasso and MLP	AUC: 0.998
Zhenguo Liu et al. [11]	2021	3D-DenseNet and Machine learning	Acc: 0.724
This study	2024	ROI and Transformer models and Feature set merging and mRMR and SVM	Acc: 1.00

## Data Availability

The original contributions presented in the study are included in the article, further inquiries can be directed to the corresponding authors.

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
