# Peer review of "Detection of Thymoma Disease Using mRMR Feature Selection and Transformer Models"

_diagnostics, 2024, doi:10.3390/diagnostics14192169_

Round 1

Reviewer 1 Report

Comments and Suggestions for Authors

General: The topic of research is original. Authors have tried different soft computing models for detection of Thymoma Disease. Latest literatures have been referred and authors have found gaps and addressed the issues very effectively. The research is in healthcare and methods used can complement different clinical methods used by doctors. May be used for cross validation. Different models have been tried and good accuracy achieved in comparison with existing methods explained in literatures.

In comparison to other published material, the proposed work shows quantitatively better results which have been explained very well with plots and confusion matrix. However, a few points need to be addressed / clarified as given below:

Abstract: Authors have explained with the performance parameters mentioned.

Methodology: Whether Images were subjected to preprocessing? If so, explain how it is done.

Results: Author claim that overall accuracy achieved is 100% / 99.22 % (cross validation). No soft computing techniques can achieve it practically. The author can explain , how this has been achieved for the interest of readers ( specific to method used / features etc if any). Other details well explained / discussed.

Conclusion: Whether authors have verified the results against pathological tests? (validation and acceptance by doctors done?)

References:   Appropriate.

No comments on the tables and figures.

No other comments on any other sections. Good work.

Author Response

Dear Reviewer #1,
We have uploaded our responses to your comments to the system as a word template. Thank you for your efforts. Best regards,

Reviewer 2 Report

Comments and Suggestions for Authors

This study proposes a novel approach for thymoma disease detection using transformer models and feature selection techniques. The approach achieves high accuracy in classifying thymoma images, demonstrating the potential of transformer models in medical image analysis. I have a few comments, detailed below.

Consider using Grad-CAM to visualize the attention areas of the transformer models, providing insights into the features that contribute most to the classification decisions. This can help understand the model's reasoning and identify potential areas for improvement.

Author split data 80-20 in training and test sets and should also use external validation set for generisability test (TCIA, kaggle or other platform-based datasets)

Author Response

Dear Reviewer #2,
We have uploaded our responses to your comments to the system as a word template. Thank you for your efforts. Best regards,

Reviewer 3 Report

Comments and Suggestions for Authors

The author must respond line-by-line to the following comments and suggestions:

1. The author(s) should avoid mentioning their contributions in the Conclusions lines 561–571.

2. Why did they call ROI, This study utilized the entire CT kernel. This is not the specific tumor section. It was to be anticipated if their model concentrated on a particular tumor ROI.

3. On line 465, mRMR ranked the features, as presented in Table 8. How did mRMR accomplish this?

4. What does it mean to rank the top/best 50, 100, and so on features? Does the scoring model assigned to each feature hold any significance? How did they select the best/top features?

6. Figure 7(d) appears stable after 2-3 epochs, a situation that is far from ideal. Although this model achieved 95.45>Deit3/base patch16>MaxViT/base-tf whereas f(a-b) is actable. Why did this happen?

7. The top 500 archived maximum results provide a vast set of features for clinical applications.

8. How will they describe which particular features were most significant for clinical application? For example, the radiomic feature could say something about tumors, I'm recommending an article https://doi.org/10.1016/j.jtho.2023.09.260 for their motivation and reference.

9. The author did not confirm that there are no common patient CTs within the training and test sets. I appreciate the robust evidence to back up my future evaluation.

Author Response

Dear Reviewer #3,
We have uploaded our responses to your comments to the system as a word template. Thank you for your efforts. Best regards,

Round 2

Reviewer 3 Report

Comments and Suggestions for Authors

The method and data distribution are not appropriate for this study. The training and test samples include a common patient. Therefore, the results look very high, and a large number of selected features have no clinical meaning. When you split a patient's CT scans between the train and test, the CT analysis consistently yields a good result. However, their response is not sufficient for me. Thanks